# Bionic e-skin with precise multi-directional droplet sliding sensing for enhanced robotic perception

Yunlong Xu [1,2,3], Zhongda Sun[2,3], Zhiqing Bai [4,5] ✉, Hua Shen[1], Run Wen[1], Fumei Wang[1], Guangbiao Xu [1,5] ✉ & Chengkuo Lee [2,3,5] ✉

Electronic skins with deep and comprehensive liquid information detection are desired to endow intelligent robotic devices with augmented perception and autonomous regulation in common droplet environments. At present, one technical limitation of electronic skins is the inability to perceive the liquid sliding information as realistically as humans and give feedback in time. To this critical challenge, in this work, a self-powered bionic droplet electronic skin is proposed by constructing an ingenious co-layer interlaced electrode network and using an overpass connection method. The bionic skin is used for droplet environment reconnaissance and converts various dynamic droplet sliding behaviors into electrical signals based on triboelectricity. More importantly, the two-dimensional sliding behavior of liquid droplets is comprehensively perceived by the e-skin and visually fed back in real-time on an indicator. Furthermore, the flow direction warning and intelligent closed-loop control of water leakage are also achieved by this e-skin, achieving the effect of human neuromodulation. This strategy compensates for the limitations of e-skin sensing droplets and greatly narrows the gap between artificial e-skins and human skins in perceiving functions.

Intelligent robots[1–3] and other smart devices[4,5], as powerful helpers for human work, have been widely used in military reconnaissance[6], rescue[7], unknown environment exploration[8], and some assistant equipment[9–12]. Environmental reconnaissance provides timely, accurate, and reliable information that strongly supports the development and implementation of strategic plans. The most fundamental aspects of intelligent robotic devices to acquire external information are perceptual modules, including various distributed sensor nodes and the integrated bionic electronic skin (e-skin)[13–16]. Among them, flexible bionic e-skin that perceives the surrounding physical information by imitating the unique function of human skin has demonstrated many compelling advantages[12,17,18].

Over the past years, a great deal of e-skins based on different working mechanisms such as piezoelectric[19], capacitive[20], and resistive[21] have been proposed to pursue perceiving capabilities closer to human skin. The proposed flexible e-skins (or skin-like sensor arrays) can detect a variety of external physical information[16,22,23], such as mechanical pressure[24], temperature[25], air humidity[26], and even finger sliding trajectories[27,28]. However, these aforementioned devices tend to only monitor the information of solid objects and air, which constrains the comprehensive perception of the droplet environment. In reality, the liquid is also the substance that intelligent devices often come into contact with or monitor in the working environment[29,30]. On the one hand, in environmental reconnaissance work for military,

[1]Key Laboratory of Textile Science & Technology, Ministry of Education, College of Textiles, Donghua University, Shanghai, China. [2]Department of Electrical & Computer Engineering, National University of Singapore, Singapore, Singapore. [3]Center for Intelligent Sensors and MEMS, National University of Singapore, Singapore, Singapore. [4]Key Laboratory of Multifunctional Nanomaterials and Smart Systems, Suzhou Institute of Nano-Tech and Nano-Bionics, Chinese Academy of Sciences, Suzhou, China. [5]These authors jointly supervised this work: Zhiqing Bai, Guangbiao Xu, Chengkuo Lee. ✉e-mail: zqbai2023@sinano.ac.cn; guangbiao_xu@dhu.edu.cn; elelc@nus.edu.sg

rescue, and scientific exploration, intelligent robotic devices need to comprehensively obtain information on weather, wind, temperature, and liquid of the environment[31]. One of the important detection factors is the falling liquid droplets in the environment (rain, dew, leaking liquid, etc.), which usually have a critical impact on the development and implementation of action plans. The most convenient and accurate way for reconnaissance robots to acquire this liquid information is to directly perceive it with a droplet e-skin (DES). Nevertheless, current e-skins are not yet able to fully sense the droplet environment and the dynamic motion behavior of the droplets. On the other hand, with the popularity of intelligent robotic devices in the future, a potential problem is that some intelligent robot workers are often vulnerable to liquid attacks in working environments, such as smart workshops, restaurants, laboratories, and smart medical rooms. Especially, it is difficult to determine the exact sliding directions and locations of liquid attack, which may bring great trouble to the safety of surrounding people, the manual operation, and the equipment maintenance. It is of great importance in environmental reconnaissance and social service equipment to develop e-skins capable of fully sensing the sliding behavior of droplets.

Droplet sensors based on liquid–solid triboelectric nanogenerators (LS-TENG) are most promising to be developed into DES because they directly convert droplet motion into electrical signals without additional power supply[32–34]. The LS-TENG is used not only for droplet energy harvesting[35,36], but also as a self-powered sensor to monitor droplet parameters by analyzing the characteristic information of electrical signals[37–39]. For example, the LS-TENG with a top electrode is developed to detect droplet parameters, such as the dropping height, dropping frequency, and other liquid properties by analyzing the unique characteristics of the output pulse of the sliding droplet[40]. Researchers also designed a cone-shaped interdigital electrode based LS-TENG by employing an interdigital electrode to increase the electrical output. This device can be used as a sensor to sensitively monitor the surface angle of the device, droplet volume rate, velocity, and droplet frequency[41]. The LS-TENG can also be used as liquid droplet counters. For instance, based on a flow-through front surface electrode and metal–dielectric junction, a self-powered water drop counter was proposed[42]. Each droplet signal generated by the drop counter can flash the LED and then be detected by a silicon photodetector, achieving precise detection for liquid drops. Besides, a self-powered microfluidic sensor based on LS-TENG was developed by utilizing the triboelectric effect of liquid droplets/bubbles inside a thin tube for real-time liquid/gas flow monitoring[43]. The flow rate and flow volume can be effectively derived based on the interval time between signals and the number of signals in a certain interval. There is also a single-electrode LS-TENG with a p-type silicon surface proposed for droplet leakage identifying and detecting[44]. This device is sensitive to leaking liquid and could qualitatively analyze liquid leakage rates. The fiber-based self-powered droplet sensor we proposed earlier can be widely used to monitor the fog intensity, the amount/frequency of raindrops, and water leakages[45]. It should be noted that all these droplet sensors can only monitor the simple or single-directional dynamic sliding information of the droplet (e.g., dropping frequency, droplet diameter, sliding velocity, and acceleration). The recently proposed droplet sensor consists of four LS-TENG units in a single-electrode mode that can monitor the position of dropping droplets at a multi-directional (two-dimensional) level, which may be a breakthrough in the perception dimension[46]. Nevertheless, the monitoring information is still not comprehensive enough, and the crosstalk appears among the multi-channel signals due to limitations in the structural design. The ideal DES should not only determine simple features for droplet contact but also perceive the more complex two-dimensional information (e.g., sliding positions, directions, and even trajectories). To this end, developing an e-skin that comprehensively senses droplet

information is expected to enhance the perception and autonomous control capabilities of intelligent robotic devices.

In this work, the self-powered bionic DES are constructed by designing co-layer interlaced branched electrode networks and using overpass connection technology. Bionic DES is composed of the upper fluorinated ethylene propylene (FEP) layer, co-layer interlaced electrode networks, and PTFE non-woven fabric substrates. With flexible materials and structures, the DES adapts well to the curved surfaces of bionic robots and other smart devices. Based on the ingenious structural design of DES, the droplet sliding behaviors can be tracked and converted into electrical signals, which facilitates the recording and transmitting of droplet information. With these advantages, the multiple kinetic parameters and multiple-directional motion behaviors of liquid droplets are successfully detected and fed back. Intelligent direction warning and closed-loop control of liquid leakages are also implemented by DES, achieving the perception and autonomous regulation like human neuromodulation.

## Results
### Design of the self-powered bionic DES
Figure 1a depicts a DES-based smart sensing system for droplet environment reconnaissance and perception inspired by the human skin. This bionic DES sensing system provides a comprehensive perception of droplet sliding, real-time feedback, and active control of liquid leakages for intelligent robotic devices. The structure and manufacturing process of the self-powered DES are depicted in Fig. 1b and Supplementary Fig. 1. The bionic DES is composed of PTFE non-woven fabric substrates, co-layer interlaced electrode networks, and the upper FEP negative triboelectric layer. In this composition, the conductive fabrics are processed into special cross shapes and are used as branched electrode units. The row electrodes and column electrodes are sequentially attached to the fabric substrates to form the co-layer interlaced X−Y electrode networks (Fig. 1b(ii, iii)). It is worth noting that the branched structure in the interlaced electrode network can nearly double the electrode coverage area as depicted in Supplementary Fig. 2, and further improve the sensing capability of DES without increasing the number of electrode channels. Furthermore, the continuous electrode units are connected by the overpass connection method (Fig. 1c). The X- and Y-electrodes interlaced with each other on the same layer without conduction. Using this connection method, the co-layer interlaced electrode networks exhibit two main advantages: there are no additional spacer layers between the X- and Y-electrodes, ensuring consistent charge induction on each electrode (see Supplementary Note 1 and Supplementary Fig. 3); in addition, this electrode configuration avoids overlapping areas of two series of electrodes and greatly reduces the signal crosstalk. As a desirable triboelectric negative material[47], the FEP film is selected to cover the surface of electrodes to complete the preparation of bionic DES. The FEP layer can be replaced at any time in subsequent applications.

The DES prepared through well-designed structures and materials meets the application requirements of intelligent robotic devices. SEM image (Fig. 1d) shows that the electrode layer and FEP triboelectric layer are visible on the PTFE fabric, and the thickness of the FEP triboelectric layer is about 0.02 mm. Besides, because DES is made of soft materials, this device exhibits good flexibility and great adaptability on non-flat surfaces such as robotic limbs (Fig. 1e and Supplementary Fig. 4). In addition, to adapt to the droplet environment, DES must also be waterproof[48]. The photograph and the water contact angle (WCA) image in Fig. 1f demonstrate that the triboelectric surface of the DES exhibits good hydrophobicity, which ensures the smooth sliding of water droplets to generate sufficient charges. Moreover, as shown in Supplementary Fig. 5 and Supplementary Movie 1, the DES remains dry even after being completely submerged and rinsed with tap water,

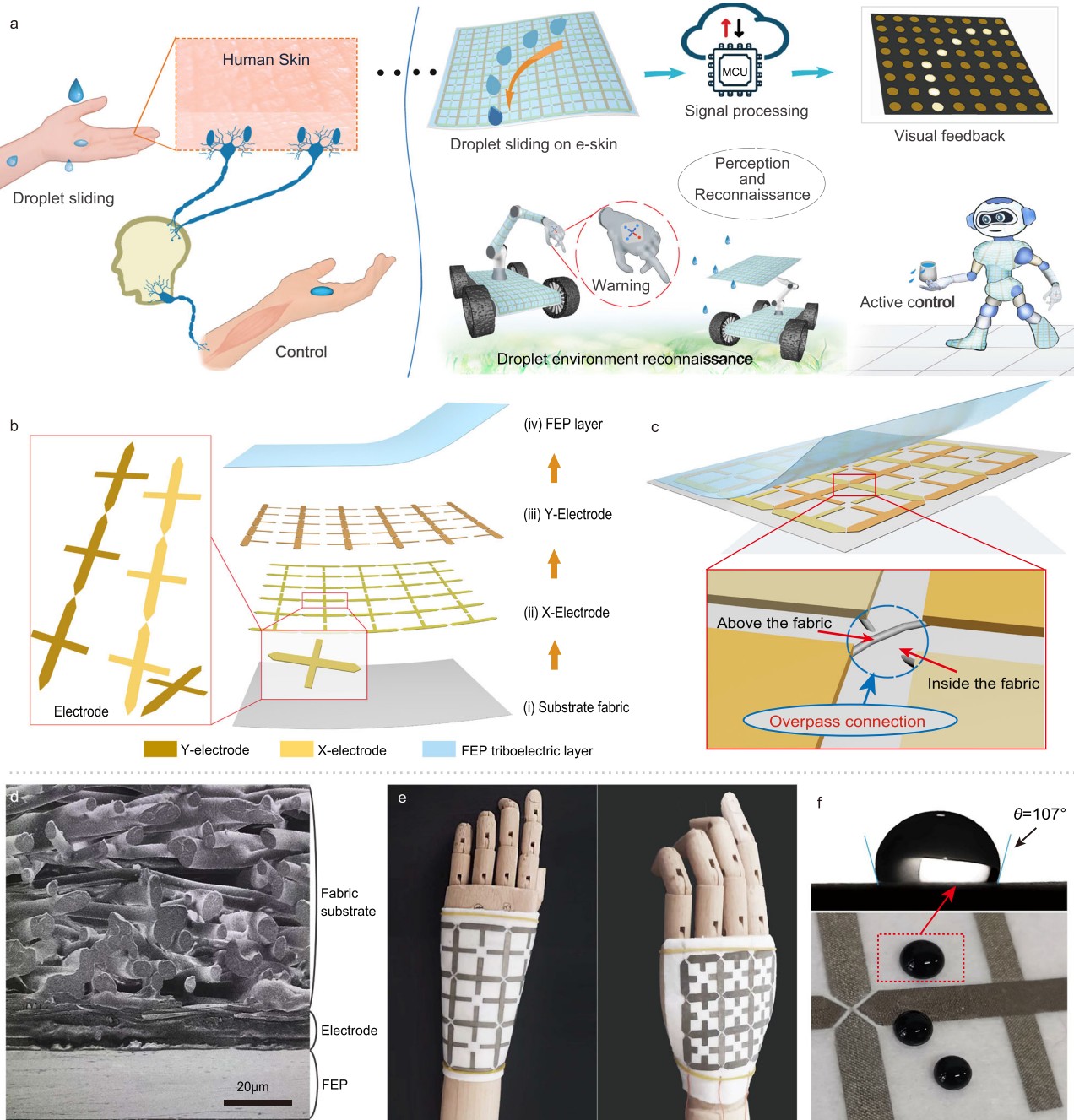

**Fig. 1 | Design of the self-powered bionic DES. a** Schematic illustration of the human tactile nervous system, and the bionic DES sensing system for droplet environment perception and reconnaissance. **b** Detailed structure of the bionic DES. The electrode unit is delicately designed as a branching structure. **c** Co-layer interlaced electrode configurations, and enlarged view of overpass connection. **d** Scanning electron microscopy (SEM) image of the cross-sectional structure of bionic DES. **e** Flexible and curved fit performance. **f** Desirable hydrophobic properties of bionic DES.

demonstrating that the DES is waterproof and suitable for applications in droplet environments.

## Working mechanism and electrical output characteristics of self-powered DES

To quantitatively investigate the ability of DES to translate the dynamic droplet motion behavior into electrical signals, a signal acquisition platform was first established as depicted in Fig. 2a. The electrical signals of the droplet sliding behavior can be acquired directly by this platform, and the mode of dual-channel or multi-channel signals can be selected according to the experimental requirements. The effective operation of DES depends on the coupling effect of friction electrification and electrostatic induction among water droplets, triboelectric layers, and electrodes[36]. Initially, due to the strong electronegativity of the FEP triboelectric layer and the generation of negative charges[49–51], the electrodes are induced and exhibit an equal number of positive charges, achieving an electrostatic equilibrium[51]. As the positively charged droplets slide toward the first electrode, the previous equilibrium is broken and a new electrostatic equilibrium between the FEP surface and the water droplet is formed (Fig. 2b(i)). To neutralize the excess positive charge in the electrodes, a stream of electrons flows from the ground to the electrodes through an external circuit, creating the electrical signal[52,53]. As depicted in Fig. 2b(ii), once the water droplet approaches the second electrode, the first electrode

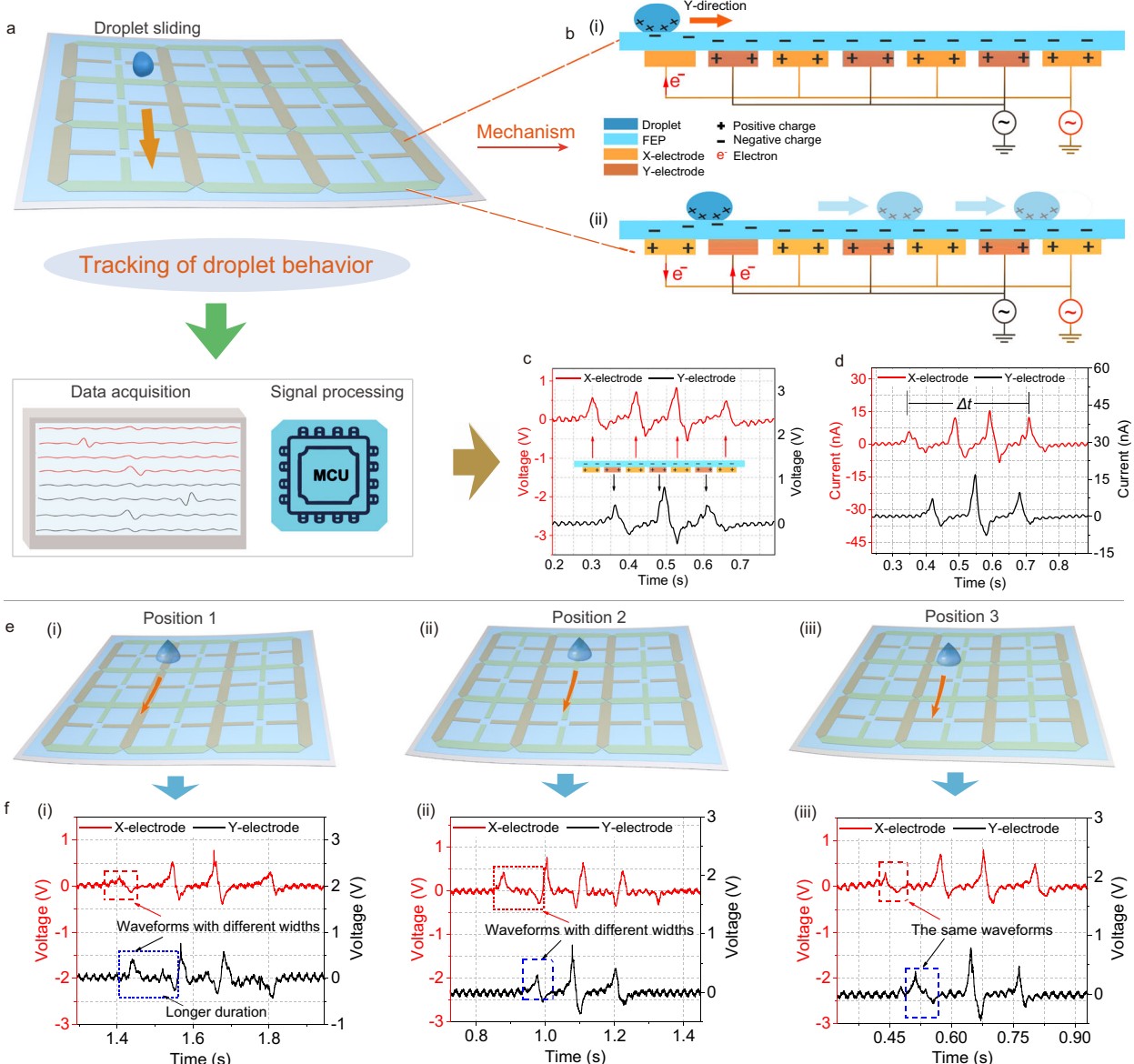

**Fig. 2 | The working mechanism and signal characteristics. a** Schematic illustration of droplet sliding behavior monitoring. **b** Schematics of the working mechanism of DES. **c**, **d** Waveforms of voltage and current generated by droplets sliding on the DES surface. **e**, **f** Droplets sliding at different three positions on DES and the corresponding output signals.

induces an opposite current, and the electron flow is induced from the ground to the second electrode[35]. Supplementary Fig. 6 shows the charge induction state of the DES as the droplet slides over the two neighboring electrodes, with the electrical signal generated throughout the row or column electrode.

The detailed dynamic sliding behaviors of droplets on the DES surface are directly converted into specific electrical signals. In this experiment, the four row electrodes are denoted as X-electrode, and the other four column electrodes are denoted as Y-electrode. Figure 2c depicts the dual-channel output voltage of about 0.85 V generated by a sliding droplet. As a water droplet slides over the surface of DES in the Y-direction, four and three electrical signal peaks are generated at the X and Y series electrodes, respectively. It is worth noting that the number and the sequence of these signal peaks are consistent with the corresponding electrodes through which the water droplets pass. Similarly, the current signal follows the same principle (Fig. 2d). More importantly, according to the characteristics of the electrical signal, the average velocity ($v$) and

average acceleration ($a$) of the sliding droplets can be derived through the following equation:

$$v = \Delta l / \Delta t \qquad (1)$$

and

$$a = 2\Delta l / \Delta t^2 \qquad (2)$$

where $\Delta l$ is the distance between four electrodes, $\Delta t$ is the time difference between the four signal peaks. As a result, it is calculated that the average velocity is 0.38 m s$^{-1}$ and the acceleration is 2.08 m s$^{-2}$.

The continuous electrode network strongly ensures gapless detection of sliding droplets in any position. Branched electrode units expand the coverage of the electrode networks and further enhance sensing capabilities (Supplementary Fig. 2). Figure 2e depicts a water droplet sliding across the four X-electrodes in the Y-direction at three different positions on the DES surface. The

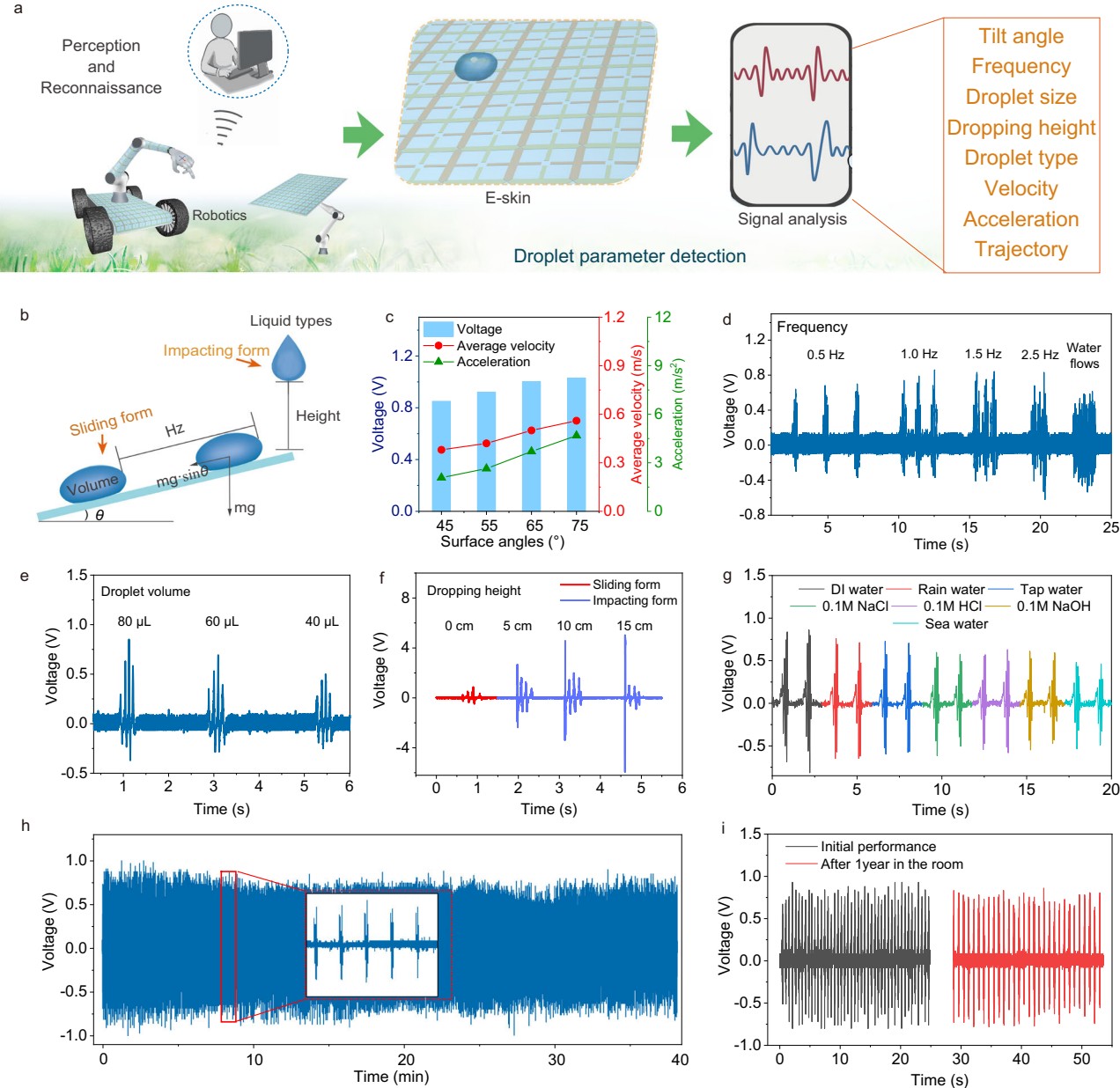

**Fig. 3 | Characterization of the sensitivity and stability of DES. a** Schematic of monitoring for droplet parameters. **b** Scheme diagram of the measurement setup. **c** Electrical output of DES at different surface angles. **d–f** Output voltages generated by water droplets with different sliding frequencies, volumes, and dripping heights. **g** Electrical signals generated by various kinds of liquid. **h, i** Continuous electrical stability and long-term electrical stability of DES.

corresponding electrical signals show that the X- and Y-electrodes, respectively, generate four signal peaks and three signal peaks, regardless of the sliding positions (Fig. 2f). This consistent result is because all the droplets passed through the X-electrodes four times and the Y-electrodes three times, which is consistent with the previous discussion. Interestingly, it can be noticed from the marked signal peaks that the wave widths of the electrical signals generated by the two serials of electrodes are slightly different. This difference in wave width is mainly due to the different contact durations between the water droplets and the electrode units along the sliding path. This finding demonstrates that the signals generated at different positions of DES are almost consistent, showing good applicability. Moreover, these slight difference in waveforms also reflects the effective discrimination for different sliding positions. To facilitate signal feature identification, we mainly chose position 3 in subsequent experiments.

## Sensitivity and stability of DES for droplet parameter detection

The excellent sensitivity of DES is essential for droplet environment reconnaissance in Fig. 3a. In this regard, we purposefully designed and conducted experiments to detect the dynamic behaviors of liquid droplets under various changing conditions. Figure 3b shows the scheme diagram of the experiment setup with different surface angles of DES, droplet volumes, sliding frequencies, dropping heights, and liquid types of the droplet. The electrical output and kinetic parameters at different surface angles of DES were first explored, as shown in Fig. 3c and Supplementary Fig. 7. As the surface angles increase from 45° to 75°, the output voltage increases significantly. In this case, the calculated average velocities of water droplets sliding are 0.38, 0.42, 0.50, and 0.56 m s$^{-1}$, and the accelerations are 2.08, 2.64, 3.70, and 4.68 m s$^{-2}$, respectively. This result shows that droplets can get higher sliding velocities at larger surface angles, thus improving the efficiency of electrostatic induction and electrical performance[54]. The finding

also demonstrates that the surface angle of DES can be detected according to the changes in electrical signals.

The detection of the parameters and motion state of the droplets themselves are also systematically performed. For instance, the sliding frequency and volumes of water droplets are important parameters that DES must detect. As shown in Fig. 3d, the electrical output shows an enhanced trend with the frequency of water droplets increasing from 0.5 to 1.5 Hz. The enhancement of electrical properties is because higher sliding frequency can compensate for the dissipation of surface charges of DES to maintain the saturation level of charges[40,45,55]. It is important to note that although higher frequencies and a continuous water flow produce a high-frequency electrical signal, the water disrupts the electrostatic induction on the electrodes, and degrades the electrical output performance. As depicted in Fig. 3e, with the volume of the water droplet decreasing from 80 to 40 μL, the electrical output decays. This change in electrical performance is mainly attributed to a sharp reduction in the contact area between the triboelectric layer and the water droplets, which decreases the generation of triboelectric charges[36,56]. In addition, Fig. 3f shows that, within a certain range, the higher the dropping height, the larger the electrical signals will be. The significant change is because the water droplet with a larger dropping distance can achieve a faster impact speed and full contact with the triboelectric layer of DES[57,58]. Moreover, larger distances lead to longer friction times between the droplet and the air, hence generating more positive charges and better electrical properties[59,60]. Although a high dropping height leads to better electrical output from the first electrode, the electrical performance decreases as the water droplets move in a sliding form on the subsequent electrodes. The experimental setup and processes of droplet detection are shown in Supplementary Fig. 8 and Supplementary Movie 2. Furthermore, different liquid types, such as deionized water (DI water), tap water, rainwater, 0.1 M NaCl, 0.1 M NaOH, 0.1 M HCl solutions, and seawater, are used as droplets for the investigation. The results show that DI water generates the highest voltage, followed by the rains and tap water (Fig. 3g). However, the other acid/alkali solutions and brines exhibited lower electrical signals, especially the lowest for seawater. The most significant reason why DES can clearly distinguish between various kinds of liquid should be the different triboelectric charges generated by them. Specifically, pure DI water does not contain excess $Ca^{2+}$, $Mg^{2+}$, $Na^+$, and $K^+$, as well as acid ions, and hence avoids the negative effects of charge-shielding, therefore generating the highest voltage[51,59,61]. The charge-shielding of high-concentration ions in ions liquid, especially seawater, reduces the charge density on the DES surface, resulting in a lower electrical output[62]. The above investigations show that the changes in external conditions can be reflected in the captured electrical signals, demonstrating the satisfactory sensitivity of DES.

In addition to the sensitivity, the desirable electrical stability of DES is also essential for its long-term operation in complex environments. The continuous and long-term electrical stability of DES are investigated in Fig. 3h, i. The electrical output signal was measured continuously for 40 min using tap water. There is a slight degradation in electrical output during the first 10 min, and no further degradation in electrical output is observed during the next 30 min of measurement. The electrical degradation should be due to the negative effects of charge-shielding caused by ions in water[63]. Furthermore, compared to the initial electrical performance, the electrical output of the DES after 1 year of indoor storage shows only negligible degradation. For resistance to mechanical deformation, the electrical output of DES before and after 1000 repetitions of 90° bends is also studied (Supplementary Fig. 9). It is easy to find that DES does not exhibit obvious signal decay after repeated deformation. These measurements strongly demonstrate the excellent electrical stability of bionic DES.

## Depth perception and visual feedback of dynamic droplet sliding by DES

To further demonstrate the ability of DES to comprehensively perceive dynamic droplet information for intelligent reconnaissance equipment, this section focuses on showing the depth perception of multi-directional dynamic sliding behaviors of droplets (i.e., trajectories). The schematic of the acquisition of multi-channel electrical signals is shown in Supplementary Fig. 10. As previously stated, the corresponding electrodes can be excited and generate electrical signals when the water droplets slide above the electrode. Specifically, a water droplet slides along the $Y$-direction following the path in Fig. 4a. The dual-channel signals show that four electrical signals appear at the X-electrodes and three electrical signals appear at the Y-electrodes (Fig. 4b(i)). These signals demonstrate that the droplet passes through the X-electrodes four times and the Y-electrodes three times. More importantly, as we can see from the multi-channel signals in Fig. 4b(ii), four electrical signals are sequentially generated at the $X_1$, $X_2$, $X_3$, and $X_4$ electrodes, respectively, while the other three electrical signals are all output from the $Y_2$ electrode in different time sequences. Depending on the electrodes and time sequence of signal generation, the trajectory of the water droplet can be accurately determined as $X_1{\rightarrow}Y_2{\rightarrow}X_2{\rightarrow}Y_2{\rightarrow}X_3{\rightarrow}Y_2{\rightarrow}X_4$, which can be derived as the trajectory map in Fig. 4c. It is worth noting that the precise position perception mainly relies on the synergistic effect of the branched electrode and the co-layer interlaced electrode networks. From the controlled experiment in Supplementary Fig. 11, it can be concluded that only the branched electrode networks can recognize the precise multiple-directional motion behavior of water droplets. As previously described, the branched structure in the electrode network can improve the sensing capability of DES by extending the electrode coverage area without increasing the number of channels. With the co-layer interlaced electrode network and overpass micro-connection, the electrode configuration avoids area overlap of two series of electrodes and greatly reduces the signal crosstalk. As shown in another control experiment in Supplementary Fig. 12, the electrical signal of the DES without the co-layer electrode and overpass micro-connection method (i.e., structure in Supplementary Fig. 3b). These signals show obvious crosstalk in both dual-channel signal and multi-channel signal. This finding directly confirms that the structural design of DES effectively improves sensing accuracy with fewer electrode channels.

The sliding behavior of droplets along the $X$-direction can also be accurately perceived. The electrical signals in Supplementary Fig. 13 show that three electrical signals are output from the $X_2$ electrode in different time sequences, and the other four electrical signals are generated sequentially on each of the Y-electrodes. Based on the previous discussion, it can be determined that the droplet slides along the trajectory of $Y_1{\rightarrow}X_2{\rightarrow}Y_2{\rightarrow}X_2{\rightarrow}Y_3{\rightarrow}X_2{\rightarrow}Y_4$ (see Supplementary Note 2). Besides, different from the movement of droplets in the $Y$- and $X$-direction, the dual-channel electrical signals of oblique sliding in Fig. 4d show that three electrical signals are generated from both Y- and X-electrode, respectively (Fig. 4e(i)). These signals mean that the droplet passes through the X- and Y-electrodes three times each. In particular, the multi-channel signals in Fig. 4e(ii) show that three electrical signals are output from the $X_2$, $X_3$, and $X_4$ electrodes, while the other three electrical signals are generated sequentially on the $Y_1$, $Y_2$, and $Y_3$ electrodes, respectively. Accordingly, it can be determined that the droplet slides along the trajectory of $Y_1{\rightarrow}X_2{\rightarrow}Y_2{\rightarrow}X_3{\rightarrow}Y_3{\rightarrow}X_4$ (Fig. 4f). The other different oblique sliding directions and their multi-channel electrical signals are investigated in Supplementary Fig. 14. Furthermore, the DES can also monitor the more complex loop sliding behavior of water droplets. As depicted in Fig. 4g, h, the loop motion of the droplets on the DES is divided into four single motions, and all these motions are converted into corresponding electrical signals.

In reality, DES may be exposed to more than one droplet at a time, so the device must be capable of sensing multiple droplets

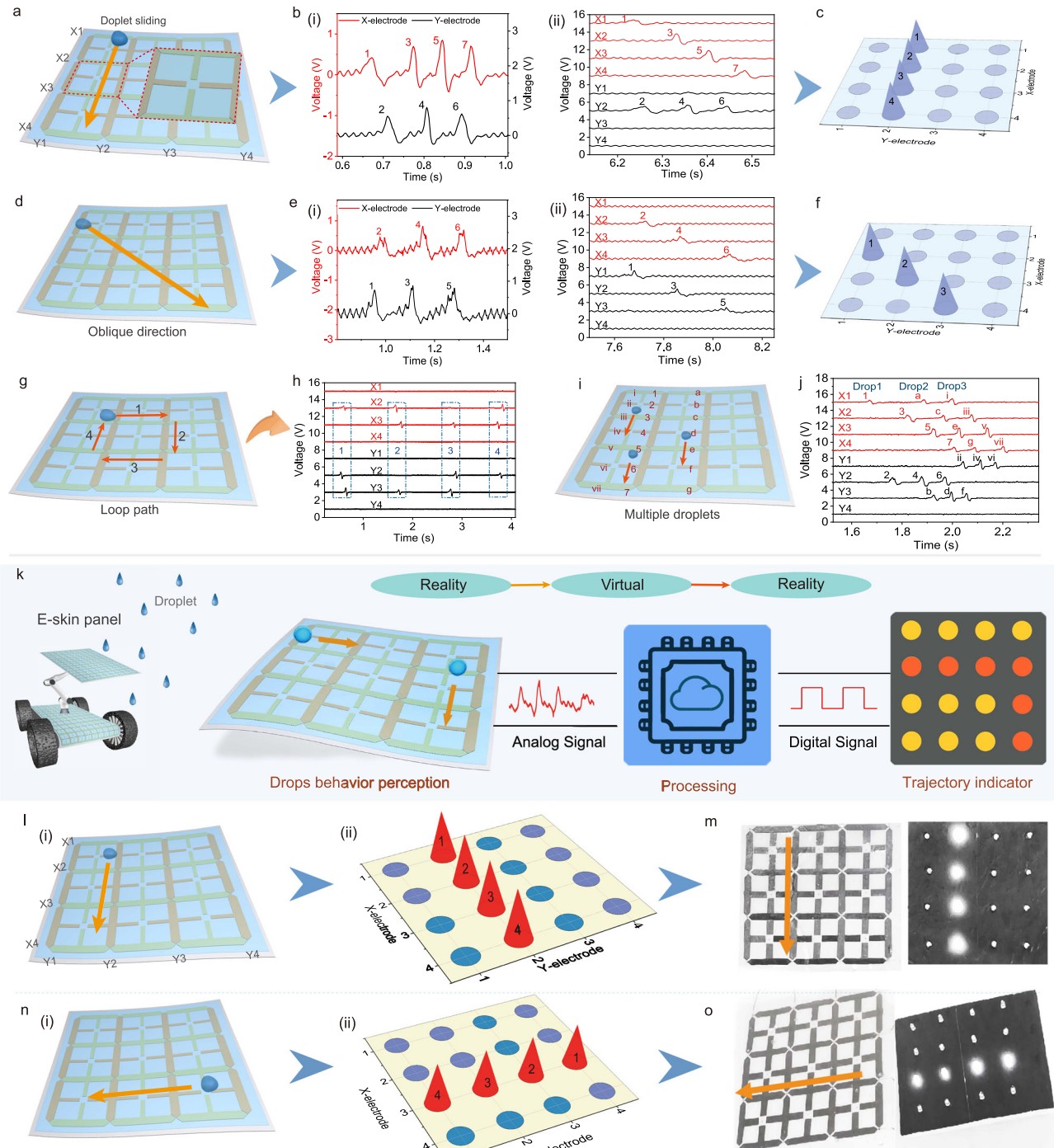

**Fig. 4 | Depth perception and visual feedback by bionic DES. a–c** Schematics of a droplet sliding in the *Y*-direction, the corresponding dual- and multi-channel output voltages, and the trajectory maps. **d–f** Droplet slides obliquely, the corresponding output signals, and the trajectory maps. **g, h** The loop sliding behavior of the droplet and the electrical signals. **i, j** Schematics of multiple droplets and the multi-channel electrical signals. **k** Dynamic droplet behavior monitoring and visual feedback system. **l** Droplet sliding in the *Y*-direction and the trajectory maps. **m** Visual feedback of droplet sliding in the *Y*-direction. **n** Droplet sliding in the *X*-direction and the trajectory maps. **o** Visual feedback of a droplet sliding in the *X*-direction.

simultaneously. As shown in Fig. 4i, j, three droplets are sliding on the DES at the same time and passing through the corresponding electrodes. The order of the electrodes passed by the three droplets and their sequence of the electrical signals are labeled as 1–2–3–4–5–6–7, a–b–c–d–e–f–g, and i–ii–iii–iv–v–vi–vii, respectively. The multi-channel electrical signals demonstrate that the sequence and position of the electrical signals generated correspond exactly to the actual path of the droplet sliding. This finding strongly confirms that the

motion behavior of these multiple droplets is still accurately perceived by DES.

To provide background workers with more direct and effective feedback on droplet sliding information, we developed a dynamic trajectory perception and visual feedback system for achieving augmented perception. As shown in Fig. 4k and Supplementary Fig. 15, the system is composed of self-powered bionic DES, MUC equipped with the preprocessing circuit, and LED trajectory indicator. The 4 × 4 LED

indicator corresponds to each electrode unit of DES, and can directly feedback on the dynamic trajectory of the droplet. This sensing system does not require the intervention of huge computers and equipment and exhibits the advantages of small size and lightweight (Supplementary Fig. 16). When the DES device is induced by the droplet movement, an analog signal is generated from the corresponding electrode of DES and sent to a microcontroller unit (MCU). The analog signal is analyzed and converted into a digital signal by the MCU and then sent to the LED indicator to display the correct motion trajectory. Specifically, as shown in Fig. 4l, with the water droplet sliding along the trajectory of $X_1 \rightarrow Y_2 \rightarrow X_2 \rightarrow Y_2 \rightarrow X_3 \rightarrow Y_2 \rightarrow X_4$ on DES, the signal characteristics of the corresponding output voltages form the trajectory maps. When the trajectory information is displayed on the LED indicator, visual feedback of the dynamic trajectory information is realized (Fig. 4m). As demonstrated in Supplementary Movie 3, this process of tracking and displaying is in real time and enables accurate identification of the dynamic behavior of water droplets. Likewise, the sliding trajectory of water droplets along the *X*-direction, oblique direction, and loop path can also be effectively tracked and displayed on the LED indicator (see Fig. 4n, o and Supplementary Movie 3). These demonstrations show that bionic DES can make it possible for robotic devices to realize the augmented perception of droplets and express their feelings about sliding droplets like human beings.

### Augmented perception and autonomic regulation of droplet sliding by intelligent robotic devices

One of the ultimate goals of in-depth monitoring for complex droplet sliding is to provide crucial information to humans. Figure 5a depicts an intelligent rescue robot that can deeply perceive the direction of a droplet sliding and send an alert to the operator. This system consists of a DES, MCU equipped with a processing circuit, LED warning light, and a robotic arm model. As a special case, the flexible DES was attached to robotic arms to perceive the droplet environment. In real applications, DES can also be integrated into other parts of the robot according to application requirements, such as the head, back, and abdomen. The perception of the liquid leakages in this demonstration is based on the collection of signal characteristics and logic analysis by the MCU. Specifically, whether the droplet slides forward, backward, or to the left on the DES, the corresponding electrical signals are induced by the DES (Fig. 5b). The signal characteristics of this electrical output can be collected by the MCU and determine the flow direction and trajectory of the droplet. At this moment, the warning light fixed on the robotic arm receives the digital signals from the MCU and then makes a corresponding warning (Supplementary Movie 4).

Furthermore, intelligent robotic devices should not only sense dynamic droplet information but also be able to take action to minimize the harm of leaking liquids. For example, in smart restaurants, service-oriented robotics must perceive liquid leakage and take timely measures to protect customers from being attacked by leaked liquid. This advanced bionic function is vividly described in Fig. 5c. The bionic closed-loop control system consists of a DES, MCU equipped with a preprocessing circuit, motor driver, stepper motor, and a hand model, as depicted in Fig. 5d(i). The detailed control mechanism is presented in Fig. 5d(ii) and Supplementary Fig. 17. Specifically, in the cup tilting and leaking scenario, the robotic hand equipped with DES perceives the liquid leakages from the cup in a specific flow direction and sends the analog signal to the MCU. Subsequently, the MCU determines the sliding direction of the leaking droplets based on logical analysis and immediately sends a digital signal command to the motor driver to adjust the robotic hand to the correct state. This perception-adjustment process is demonstrated in Supplementary Movie 5. During this process, the MCU can simultaneously send control commands to the driver and receive constant feedback from the DES. The MCU will only stop sending commands when the hand is in the correct state, thus realizing closed-loop control. This demonstration can not only realize the perception-feedback of droplet sliding behavior but also achieve autonomous regulation like humans, showing great value and potential for intelligent robotics applications.

## Discussion

In summary, to improve the augmented perception and autonomous regulation capabilities of intelligent robotic devices for liquid substances, we developed a self-powered bionic DES. The DES is developed based on triboelectric mechanism and by designing co-layer interlaced electrode networks and using overpass connection technology. This method exhibits two main advantages: the co-layer interlaced electrode network ensures consistent charge induction between the triboelectric layer surface and each electrode; this electrode configuration avoids overlapping areas of two series of electrodes and greatly reduces the signal crosstalk. As the components of DES (hydrophobic fabric substrate, triboelectric layer, and electrodes) are all made of soft materials, the DES is flexible and very suitable for attaching to non-planar surfaces of robotic devices. Based on the ingenious design, a variety of complex motion behaviors of water droplets can be converted into electrical signals, thus promoting the recording, transmission, and analysis of droplet motion information. With these advantages, we successfully demonstrated the detection of basic droplet characteristics (frequency, volume, height, and liquid type) and single-directional kinematic parameters (velocity and acceleration) by using DES. In particular, the accurate perception and feedback of multi-directional (two-dimensional) dynamic sliding trajectories of droplets are achieved. Furthermore, an intelligent flow direction warning system and closed-loop control system for droplet leakages inspired by human neuromodulation are also realized. This bionic DES is expected to greatly narrow the gap between artificial e-skin and human skin in perception ability, and bring promising applications in military, rescue, and daily life.

## Methods

### Fabrication of the flexible DES

The DES consists of a PTFE non-woven fabric substrate, electrode network, and negative triboelectric layer. Before the fabrication of DES, non-woven fabric was immersed in ethanol for 30 min, and then washed with DI water and dried. The branched electrode units are made of double-sided adhesive copper–nickel conductive fabric. The conductive fabric was processed into a special cross-shaped structure by template cutting method.

Before attaching the electrodes, we insert a copper wire in the correct position through the interior of the substrate to form an overpass connection together with the wires on the substrate surface. Subsequently, as shown in Supplementary Fig. 1, the row electrode units were first attached to the fabric substrates with the designed rules as X-electrodes. The co-layered interlaced electrode network was then formed after the column electrode units were sequentially attached as Y-electrodes. The continuous electrode units are connected by the overpass connection method. As a negative triboelectric layer, the FEP copolymer was attached to the adhesive electrodes to obtain the designed DES ($13 \times 13$ cm). Besides, other sizes of DES ($10 \times 10$ and $7 \times 7$ cm) were prepared for different application scenarios.

### Characterization

The cross-sectional microscopic architectures of DES were characterized using the scanning electron microscope (Hitachi SU8010, Japan). The WCA of the DES surface was measured by a Contact Angle Analyzer (OCA15EC, Germany). Water resistance was characterized by submerging the DES in water and rinsing it with tap water continuously.

### Electrical measurement

For the electrical measurement, DI water was used as the liquid droplets, and the droplet volume, surface angle of the DES device, and

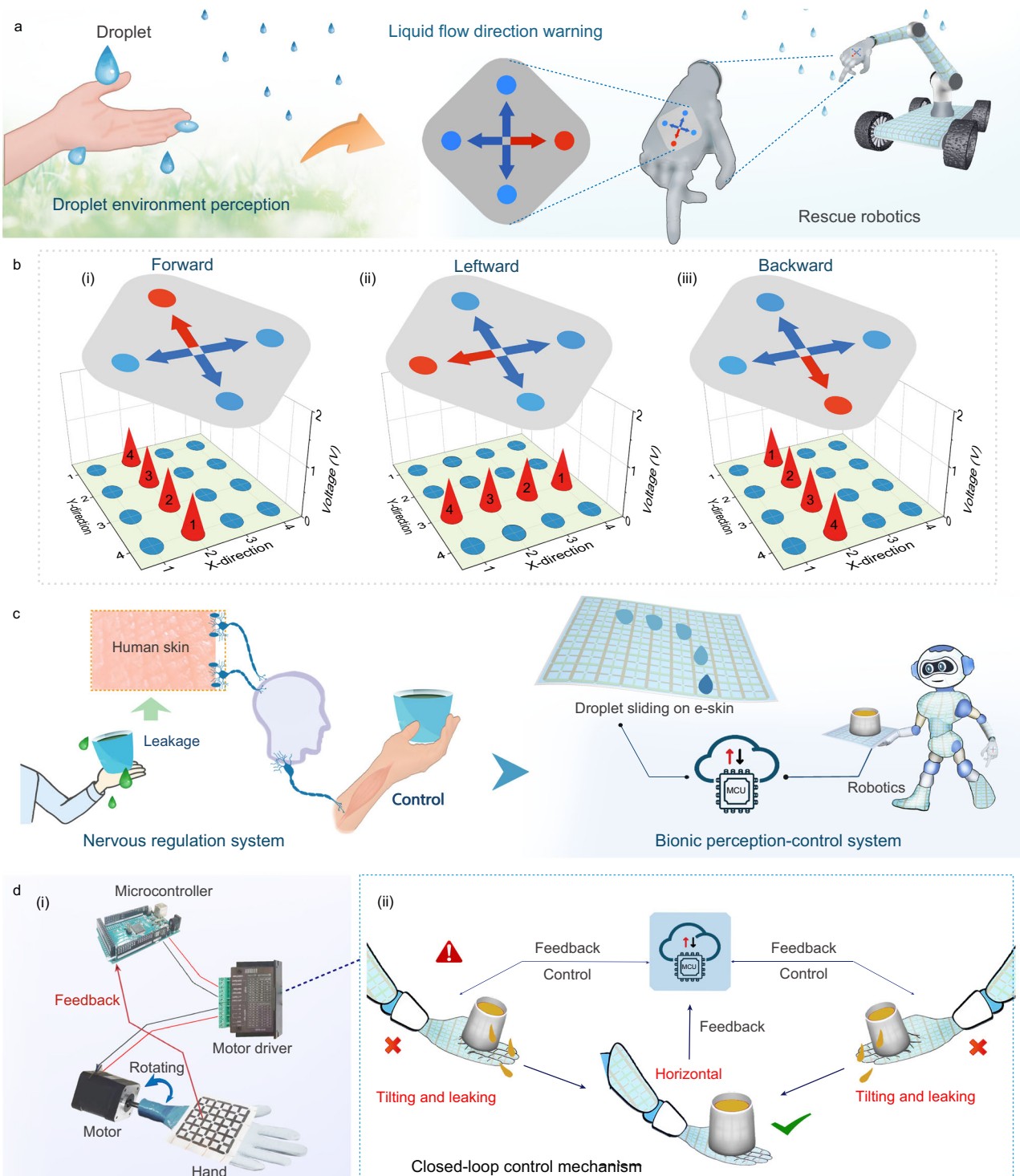

**Fig. 5 | Augmented perception and autonomic regulation for droplet sliding.**
**a** Schematic of a DES-equipped bionic rescue robot inspired by human skin for detecting the droplet environment. **b** Trajectory maps of droplet sliding and the corresponding directional warning. **c** Schematics of DES-based bionic sense-control system inspired by human neuromodulation. **d** Demonstration of an intelligent control system and closed-loop control mechanism for cup leakage in smart restaurants.

sliding frequency were set at 80 μL, 45°, and 1 Hz, respectively. The output voltages were acquired by multi-channel virtual oscilloscopes, and the output currents were collected using a programmable electrometer (Keithley 6514, USA).

For droplet parameter detection, a piece of DES is attached to the hand model and connected to a small signal indicator for real-time display of the droplet parameters. In this experiment, the inclination angle of the DES surface was controlled by using an angle plate; volumes of droplets were provided by using the droppers with different diameters; the frequency of the droplets was controlled according to the MCU frequency indicator; and the height of the droplets is measured by the height ruler. For different liquid detections, the FEP layer was replaced after each measurement. In the experiment, Singapore tap water was

used, and the rainwater was taken from the March rainfall in Singapore.

For analyzing droplet trajectories from multi-channel signals, the signal peaks should first be labeled with the correct sequence numbers based on the order in which the peaks appear in the multi-channel signal. Then read the electrode number corresponding to the electrical signal according to the labeled serial number. By listing the corresponding electrode numbers in time order, the specific trajectories of the droplet can be obtained.

### Characterization of augmented perception and autonomic regulation

In the application experiments, the Arduino MEGA 2560 MCU was used. For the motion trajectory feedback and flow direction alarm system, the LED indicator is made by integrating 16 or 4 LEDs onto the flat boards. The 4 × 4 LED array corresponds to the intersection of X- and Y-electrodes in DES. The components and connections are shown in Supplementary Fig. 16. For the closed-loop control demonstration, an MCU, a motor driver, a stepper motor, and a hand model were used. The robotic hand model equipped with DES was connected to the MCU, motor driver, and stepper motor to form a closed-loop control system. The motor and robotic hand are controlled by the MCU by accepting the sensing signals from the DES. At the same time, the state of the hand model is fed back to the MCU in real time by the DES to form a closed-loop control.

## Data availability
The authors declare that all data supporting the results of this study are present in the paper, and the data sources are available under https://doi.org/10.6084/m9.figshare.26132239.

## Code availability
The codes for microcontroller control are openly available under https://doi.org/10.6084/m9.figshare.26132029.

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

## Acknowledgements

This work was supported by National Key R&D Program of China, China [Grant Number 2022YFC3006100, G.X.]; Fundamental Research Funds for the Central Universities and Graduate Student Innovation Fund of Donghua University, China [Grant Number CUSF-DH-D-2022024, Y.X.]; the fellowship from the China Scholarship Council (CSC), China [Grant Number 202306630057, Y.X.]; the NRF-CRP28-2022-0038 "Integrating Wideband Tuneable Acoustic Filters on Silicon for High-Speed Wireless Communication" [WBS: Grant Number A-8001503-00-00, C.L.] at National University of Singapore, Singapore, and RIE2025 IAFICP under 12301E0027 "Piezo Specialty Lab-in-Fab 2.0 (LiF 2.0)-Enabling Unrivalled Power Efficient Transducers Beyond Material Limits" at National University of Singapore, Singapore, [C.L.]; the fellowship of China National Postdoctoral Program for Innovative Talents, China [Grant Number BX20240408, Z.B.], and Jiangsu Funding Program for Excellent Postdoctoral Talent, China, [Z.B.].

## Author contributions

Y.X., Z.B., and G.X. conceived the idea. Y.X. planned and performed the experiments. Y.X. took all the photos shown in the figures. Y.X. and Z.S. programmed the MCU for the demonstration and prepared the arm model and LED display. Y.X., Z.B., and H.S. contributed to the data analysis and drafted the manuscript. Y.X., Z.S., Z.B., R.W., F.W., C.L., and G.X. edited the manuscript. Z.B., C.L., and G.X. supervised the work.

## Competing interests

The authors declare no competing interests.
