## [Peer Review File · Nature Communications]

Bionic E-skin with Precise Multi-Directional Droplet Sliding Sensing for Enhanced Robotic PerceptionReviewers' comments:

Reviewer #1 (Remarks to the Author):

In this manuscript, an interactive bionic droplet electronic skin (DES) was developed using triboelectricity technology. The co-layer interlaced electrode network with branched electrode structure was designed. The authors suggest that the electric output is sensitive to several droplet characteristics and motion behaviors. Furthermore, two application cases were presented to show the potential functions of the designed sensor. Although the idea is interesting, there are still some major flaws that should be clarified, in terms of ambiguous mechanism and insufficient verification, to improve the originality and significance of this work to meet the high criteria of Nature Communications. Therefore, I can not recommend its publication at the present state. The following points are major issues about this manuscript.

1 The overpass connection technology should be described in detail. The author mentions that this technology can reduce signal crosstalk. Experimental data supporting the author's claim should be provided.

2 The author investigated the effects of tilt angle, dropping height, droplet volume, and frequency on the triboelectric signals. However, the manuscript does not provide clear descriptions of how these variables were precisely adjusted, and it is also not evident from Supplementary Movie 2 whether other variables remained constant when altering a specific parameter.

3 Moreover, regarding the impact of these variables on the triboelectric signals, the author should conduct further experiments or simulation analyses to delve into their underlying mechanisms. For instance, concerning the influence of frequency, there should be more mechanistic explanations. Similarly, regarding the effect of dropping height, height not only affects velocity but also influences the time the droplet spends in the air, consequently leading to changes in its charge accumulation.

4 What is the significance of tracking droplet trajectories in reality? Furthermore, for precise droplet position identification, it's apparent that more electrode units are required. However, with more electrode units, the distance between electrodes decreases, potentially causing droplets to cover multiple nodes simultaneously, thus affecting recognition accuracy. Additionally, more electrode units increase the difficulty and cost of signal acquisition.

5 The closed-loop control described in Figure 5d is unclear and requires more detailed explanation. Additional details should be provided. In practical scenarios, leaked liquid may not always be in the form of droplets, and the correspondence between the direction of droplet motion and the adjustment of the robot's orientation can be complex. This complexity arises because, in reality, the direction of droplet sliding may not simply correspond to the direction of cup tilting. Moreover, from the figures in the paper and Supplementary Movie 5, it is not apparent how the liquid leakage occurs.

6 From the manuscript, it is evident that the signals generated by the DES are relatively weak, typically in the range of tens of nA with voltages around 1 V. In robot applications, signal fluctuations resulting from electromagnetic interference produced by the robot's motors, drivers, and other components could potentially overshadow the signals generated by the droplets. However, the authors have not described how these issues are addressed in the manuscript.

7 The authors claim that the designed DES exhibits excellent flexibility. However, the manuscript does not provide information on the electrical output of the DES under deformation or the stability of the signals during repeated deformation processes. Additionally, the application cases presented in the manuscript show that the DES is fixed on a flat surface. In reality, however, the surfaces of robotic hands or human skin are not flat.

Reviewer #2 (Remarks to the Author):

This manuscript reported an electronic skin capable of tracking the two-dimensional dynamic sliding trajectories of water droplets. The electronic skin consists of a nonwoven fabric substrate, flexible interlaced electrode, and a negative triboelectric film, showing good flexibility and conformity to non-planar surfaces such as robot limbs. The key novelty of the electronic skin lies in the interlaced electrode design which enables the detection of temporal differences among the electrical signals induced by a sliding droplet and thereby allows real-time record of the droplet's motion direction. Combined with the microcontroller unit for signal processing, the electronic skin can be further utilized for intelligent closed-loop control of robots to prevent liquid droplet leakages. However, this work primarily focused on tracking the sliding motion of the individual droplet and did not address the detection of more common scenarios involving multiple droplets

sliding simultaneously. The practical implementation of this work seems to be limited due to its sole reliance on water droplets as the sensing object. Additionally, the manuscript does not provide sufficient information regarding the stability of the electronic skin's performance. I also recommend that the authors carefully reexamine their calculation results for droplet acceleration, specifically focusing on cases where the calculated values appear unreasonably high. Thus, this manuscript could be considered for publication after the authors address the following technical comments.

1. The illustration of working mechanism in Fig. 2a-d appears to be elusive. It is not easy to understand the process of electrical induction and the charge transfer induced by droplet sliding. I recommend the authors to make additional explanations in both schematics and discussions.
2. The authors chose the tap water, rain water, and DI water as the research objects in the manuscript. What is the sensing capability of the electronic skin for aqueous solutions (such as brine, acid or alkali) and organic liquids (such as ethanol or acetone)? They are more common in the environments requesting the robot workers, for example, workshops, chemical laboratories, and smart medical rooms.
3. The electronic skin can effectively sense the dynamic parameters of the single droplet. What is the electronic skin's performance in detecting multiple droplets?
4. The performance stability of the electronic skin is rarely discussed in the manuscript. Many works have mentioned that the performance of triboelectric films is vulnerable to the ions (10.1002/adma.201905696; 10.1016/j.xinn.2022.100301). Even in DI water, there exist hydrogen ions and hydroxide ions. I am curious about whether ions would undermine the stability of the electronic skin's performance and cause the transition of the sensing signals when the electronic skin experiences long term exposure to in water. In addition, would other environmental conditions, including humidity, temperature and air pressure, affect the signals?
5. On page 10, the authors claimed that the acceleration of the sliding droplet could be calculated as up to 11.87 m/s² via the signal treatment. This value has been far higher than the gravitational acceleration of 9.8 m/s². The result is confusing because the gravitation should be the main driving force to push the droplet sliding and no other force could further accelerate the sliding droplet.
6. The sensing experiments appear to set up on the static electronic skin. What about the sensing result for the electronic skin attached on a moving robotic arm?
7. To better the significance of this work, a comprehensive overview of the existing state-of-the-art droplet sensing and energy harvesting technologies should be supplemented in the introduction section of the manuscript. For example, 1. doi.org/10.1002/dro2.77.
8. The authors should provide more detailed discussions about the device fabrication. For example, how to make the cross shape of the electrode? How to firmly attach the FEP film onto the electrode surface? Is there adhesive between them? It is hard to find something there in the SEM image (Fig. 1d).
9. The contact hysteresis of the FEP surface should be provided to characterize the droplet sliding dynamic.
10. The signal treatment methods used for analyzing and sorting the droplet-induced multiple signals, particularly for cases depicted in Fig. 4b and e, are indeed crucial and require detailed discussion in the Method.
11. The results of sensing oblique and loop sliding should be arranged as main figures instead of the Supplementary figures because they are more convincing. The authors should provide more actual photos to introduce the close-loop control system and related experiments given that Fig. 5 is currently filled with schematics.

Point-by-Point Response to the Reviewers' Comments

Reply to reviewer:

Dear reviewers:

Thank you very much for the valuable comments and helpful suggestions. We have studied the comments carefully and made the correction. **The main corrections have been highlighted with yellow shading** in the revised manuscript and the point-to-point reply to the reviewer's comments are as following:

Reviewer #1 (Remarks to the Author):

In this manuscript, an interactive bionic droplet electronic skin (DES) was developed using triboelectricity technology. The co-layer interlaced electrode network with branched electrode structure was designed. The authors suggests that the electric output is sensitive to several droplet characteristics and motion behaviors. Furthermore, two application cases were presented to show the potential functions of the designed sensor. Although the idea is interesting, there are still some major flaws should be clarified, in terms of ambiguous mechanism and insufficient verification, to improve the originality and significance of this work to meet the high criteria of Nature Communications. Therefore, I can not recommend its publication at the present state. The following points are major issues about this manuscript.

Reply to reviewer:

We are very grateful to the reviewers for their comments on improving the quality of our manuscripts. We have supplemented more experimental details and characterizations (such as crosstalk validation, water flow monitoring, factor regulation methods, and detailed influence mechanisms of various factors) to improve the quality of this manuscript. We have also supplemented the mechanisms for electrical signal generation, crosstalk reduction, effects of different droplet parameters on the electrical signal, and the sense-control applications in the manuscript. In this work, we aim to develop a satisfactory droplet e-skin based on droplet sensors to meet the comprehensive sensing needs of intelligent reconnaissance robots and service robots. The current e-skin is limited to the perception of solid objects and air (ref 20-30 in manuscript), which severely constrains the comprehensive perception of the droplet environment by intelligent robotic devices. The previously reported droplet sensors can only detect simple dynamic motion of droplets in a single direction, such as dropping frequency, sliding velocity, and acceleration (ref 39-47 in manuscript). In this case, our research (DES) achieves not only the detection of basic droplet sliding information but also the in-depth perception and feedback of dynamic sliding trajectory in multiple directions (two-dimensional). The DES fills a gap in the lack of liquid droplet e-skins and narrows the gap between artificial e-skins and human skins in perceiving functions, which is certainly great progress in the sensing field. **The following is our point-to-point reply along with reviewer's comments.**

1 The overpass connection technology should be described in detail. The author mentions that this technology can reduce signal crosstalk. Experimental data

supporting the author's claim should be provided.

Reply to reviewer:

We appreciate the reviewer's valuable comment, and we are sorry that the explanation about the overpass connection technology was insufficient.

We provided a detailed description of the overpass connection technology on page 7 and page 24 in the revised manuscript. As shown in Fig. 1c, in the co-layer interlaced electrode networks, X-electrodes and Y-electrodes are in the same layer on the fabric substrate, and the continuous electrode serials are connected by overpass connection technology. The X-electrodes and Y-electrodes are interlaced but not conducting. Before attaching the electrodes, we first insert a metal wire in the correct position through the interior of the substrate to form an overpass connection together with the surface wires. As discussed on page 7 and page 18 in the revised manuscript, using the overpass connection technology, the co-layer interlaced branched electrode networks exhibit two main advantages: There are no additional spacer layers between the X-electrodes and Y-electrodes, ensuring consistent charge induction on each electrode (see Supplementary Fig. 3); In addition, this electrode configuration avoids area overlap of two series of electrodes and greatly reduces the signal crosstalk. Moreover, the narrow electrode units can also reduce the interference caused by proximity to each other.

Based on the reviewer's suggestion, we provided additional experiments in Supplementary Fig. 12 and confirmed that the overpass connection technology can reduce signal crosstalk. Supplementary Fig. 12 shows the electrical signal output of the DES without the overpass connection technology (using the structure in Supplementary Fig. 3b). The results show obvious crosstalk in both dual-channel signals and multichannel signals. In comparison, all the electrical signals of the DES with co-layer electrodes throughout the manuscript show no crosstalk problems.

Fig. 1c | Co-layer interlaced electrode configurations, and enlarged view of overpass connection structure.

Supplementary Fig. 3 | Comparison of co-layer electrode (a) and non-co-layer electrode (b). The co-layer electrode with more consistent and thinner triboelectric layers.

Supplementary Fig. 12 | The crosstalk validation of the DES without the co-layer interlaced electrode and overpass connection. (a) Schematic diagram of a droplet sliding in the X-direction, (b) the dual-channel signal and (c) the multichannel electrical signal all show obvious crosstalk.

2 The author investigated the effects of tilt angle, dropping height, droplet volume, and frequency on the triboelectric signals. However, the manuscript does not provide clear descriptions of how these variables were precisely adjusted, and it is also not evident from Supplementary Movie 2 whether other variables remained constant when altering a specific parameter.

Reply to reviewer:

We thank the reviewer for the careful review and comments, and we are sorry the explanation about the experimental conditions was insufficient.

To help readers better understand the investigations, we provided photos of the experimental setup in Supplementary Fig. 8. In this experiment, we strictly controlled the experimental parameters to achieve accurate monitoring of droplet sliding behavior by DES. In Fig. 3, we investigated the effects of tilt angle, liquid types, droplet volume, frequency and dropping height on the triboelectric signals. Supplementary Fig. 8 clearly shows the detailed setup of this experiment. We control the inclination angle of the DES surface by using an angle plate; provide different volumes of droplets by using droppers with different diameters; control the frequency of the droplets according to

the MCU frequency indicator; and the height of the droplets is measured by the height ruler. For different liquid detections, the FEP layer was replaced after each measurement. The operation details were supplemented in Method on page 25 of the revised manuscript. We believe the experimental details can be better understood with the additional photos.

Supplementary Fig. 8 | **a** Measurement platform for droplet parameter detection. **b** Height scale ruler. **c** Droppers with different diameters. **d** Angle plate. **e** MCU frequency indicator light. **f** Different liquid types.

3 Moreover, regarding the impact of these variables on the triboelectric signals, the author should conduct further experiments or simulation analyses to delve into their underlying mechanisms. For instance, concerning the influence of frequency, there should be more mechanistic explanations. Similarly, regarding the effect of dropping height, height not only affects velocity but also influences the time the droplet spends in the air, consequently leading to changes in its charge accumulation.

Reply to reviewer:

We thank the reviewer for the careful review and comments.

We agree with the reviewer that the height of droplets affects not only the impacting velocity but also the time that the droplet spends in the air, leading to changes in its charge accumulation. As described in related research (ref 59-62 in manuscript), on the one hand, larger droplet distances lead to greater electrical output because the water droplet can get a faster impacting speed and further full contact with the triboelectric layer. On the other hand, larger distances cause longer friction times between the droplet and the air, hence generating more charge and exhibiting better electrical properties.

Therefore, in the case of Fig. 3f, the height of droplet affects not only the impacting speed but also the time that the droplet spends in the air, leading to larger charges and electrical outputs. For the case of sliding frequency of droplets, within a certain range, increasing the frequency will enhance the electrical output. This is because the increase of sliding frequency limits the escape of charges on the tribomaterials surface and compensates for the dissipation of surface charges of DES to maintain the saturation level of charges (ref 42, 47, and 57 in manuscript). All the mechanisms of the water droplet parameters on the electrical output of DES are supplemented **on pages 13-14** in the manuscript as follows:

“As shown in **Fig. 3d**, the electrical output shows an enhanced trend with the frequency of water droplets increasing from 0.5 Hz to 1.5 Hz. **The enhancement of electrical properties is because higher sliding frequency can limit the escape of charges and compensate for the dissipation of surface charges of DES to maintain the saturation level of charges.** It is important to note that although higher frequencies and a continuous water flow produce a high-frequency electrical signal, the water disrupts the electrostatic induction on the electrodes, and degrades the electrical output performance. As the volume of the water droplet decreases from 80 μL to 40 μL , the electrical output decays (**Fig. 3e**). This change in electrical performance is mainly attributed to a sharp reduction in the contact area between the triboelectric layer and the water droplets, which decreases the generation of triboelectric charges and electrical output. In addition, **Fig. 3f** depicts that, within a certain range, the higher the droplet height, the larger the electrical signals will be. **The significant enhancement is because the water droplet with a larger dropping distance can achieve a faster impacting speed, and further full contact with the triboelectric layer of DES.** Moreover, larger distances lead to longer friction times between the droplet and the air, hence generating more charges and better electrical properties. Although a high dropping height leads to a high electrical output from the first electrode, the electrical performance decreases as the water droplets move in a sliding form on the subsequent electrodes. The experimental setup and processes of droplet detection are recorded in Supplementary Fig. 8 and Supplementary Video 2. Furthermore, different liquid types, such as deionized water (DI water), tap water, rainwater, 0.1 M NaCl, 0.1 M NaOH, 0.1 M HCl solutions and seawater, are used as droplets for the investigation. The electrical output shows that DI water generates the highest voltage, followed by the rains and tap water (**Fig. 3g**). However, the other acid/alkali solutions and brines exhibited lower electrical signals, especially the lowest for seawater. The most significant reason why DES can clearly distinguish between various kinds of liquid should be the different triboelectric charges generated by them. Specifically, DI water is pure and does not contain excess Ca^{2+} , Mg^{2+} , Na^+ , and K^+ , as well as acid ions that avoid the negative effects of charge-shielding, and therefore generate the highest voltage. The charge-shielding of high concentration ions in other liquid, especially the seawater, reduces the charge density on the DES surface, resulting in a lower electrical output.”

4 What is the significance of tracking droplet trajectories in reality? Furthermore, for precise droplet position identification, it's apparent that more electrode units

are required. However, with more electrode units, the distance between electrodes decreases, potentially causing droplets to cover multiple nodes simultaneously, thus affecting recognition accuracy. Additionally, more electrode units increase the difficulty and cost of signal acquisition.

Reply to reviewer:

We thank the reviewer for the careful review and comment.

In reality, liquid are the important substances that intelligent devices often come into contact with or monitor in the working environment. On the one hand, in environmental reconnaissance work for military, rescue and scientific exploration, intelligent robotic devices need to comprehensively detect the terrain, weather, wind, temperature and water of the environment. **One of the important detection factors is the falling droplets in the environment (rain, dew, leaking liquid, etc.). Droplet sliding information (e.g., sliding velocity, acceleration, direction, position, and even trajectory) has important impacts on the development and implementation of military and rescue programs.** The most convenient and accurate way to detect this liquid information is to directly perceive it with droplet an e-skin. Nevertheless, current e-skins are not yet able to sense the droplet environment and the dynamic motion behavior of the droplets. On the other hand, with the popularity of intelligent robotic equipment in the future, a potential problem is that some special intelligent robotic workers are often vulnerable to liquid attack in many working environments, such as smart workshops, restaurants, laboratories, and smart medical rooms. **It is difficult to determine the exact directions and locations of liquid attack, which brings great trouble to the safety of surrounding people, manual operation and equipment maintenance.** Moreover, current e-skins are not yet able to detect the motion behavior of surface droplets in-depth and constrain the ability of intelligent devices to comprehensively perceive real environments. It is of great importance in environmental reconnaissance and social service equipment to develop e-skins capable of fully sensing the sliding behavior of liquids.

Furthermore, the electrode spacing does decrease at high electrode densities, but the electrode density should not increase infinitely. **Excessive electrode density will cause significant crosstalk.** It is best to keep the electrode spacing just close to the diameter of the general droplets, which can not only ensure good accuracy but also prevent the occurrence of crosstalk. We have considered the contact area and diameter of the surface droplets when designing the DES, and prepared the DES devices in three sizes of 13×13, 10×10, and 7×7 cm, respectively, for different applications. The following Figure shows the state of coverage of water droplets (0.1 ml) on the surfaces of DES with different sizes. The diameter of the water droplets on one of the 7×7 cm DES surfaces is just close to the spacing between two neighboring electrodes, which is enough to sense the sliding behavior of droplets.

(a) DES with different sizes. (b) State of coverage of 0.1 ml droplets on the different DES surfaces.

5 The closed-loop control described in Figure 5d is unclear and requires more detailed explanation. Additional details should be provided. In practical scenarios, leaked liquid may not always be in the form of droplets, and the correspondence between the direction of droplet motion and the adjustment of the robot's orientation can be complex. This complexity arises because, in reality, the direction of droplet sliding may not simply correspond to the direction of cup tilting. Moreover, from the figures in the paper and Supplementary Movie 5, it is not apparent how the liquid leakage occurs.

Reply to reviewer:

We thank the reviewer's valuable comments and suggestions.

We are sorry that the explanation about the closed-loop control was insufficient. To help readers better understand the control mechanism, we have provided detailed descriptions and the photographs in Fig. 5d and Supplementary Fig. 17. We've also supplemented the detailed explanation to Supplementary Video 5 to explain how this system works. The closed-loop control system consists of DES, MCU equipped with a processing circuit, motor driver, stepper motor, and a hand model. In case of cup tilt and leakage, the DES-equipped robotic hand senses the flowing liquid in a specific direction and sends the analog signal to the MCU. Subsequently, the MCU determines the flow direction of the leaked droplets based on a logical analysis of signal characteristics and immediately sends a digital signal command to the motor driver of the robotic hand to adjust the cup to the correct state (Supplementary Video 5). During this process, the MCU can simultaneously send control commands to the driver and receive constant feedback from the DES. The MCU will only stop sending commands to the driver when the hand has been adjusted to the correct position, thus realizing closed-loop control. This detailed description of the control mechanism has also been provided on page 23 of the revised manuscript.

The demonstration in Fig. 5d is designed to show the ability of an intelligent robotic waiter to take timely action to control water leakage and avoid liquid attacks on customers in a smart restaurant. In the cup tilting and leaking scenario, the direction of droplet sliding corresponds to the direction of the cup tilting. This demo was also designed with real-time feedback in mind, i.e., it will continue to be adjusted until the end of the leakage process. For more complex situations such as cup collision and splashing, it is necessary to cooperate with the view system to identify it. Just like humans, both touch and vision are needed to determine the location of splashed droplets. We agree with the reviewer's opinion that leaked liquid may not always be in the form of droplets. In addition to the droplets, the DES is also capable of monitoring **water flows** as shown in Fig. 3d. In reality, there are indeed very complex droplet slides. For example, we explored the effective sensing of droplet lateral, longitudinal, oblique, and complex loop motions by DES in Fig. 4. These typical cases confirm that DES can sense a variety of droplet slides in reality. Moreover, As can be seen from Video 5, water leakage is achieved by filling the tilted cup with water so that it overflows. This is designed to demonstrate continued feedback and adjustment until the leak is over.

Fig. 5d | (i) Intelligent sense-control system for cup leakage and (ii) the closed-loop control mechanism used for an intelligent robot in a smart restaurant.

Supplementary Fig. 17 | Components and mechanisms of closed-loop control systems.

Fig. 3d | Output voltages generated by water droplets with different frequencies.

6 From the manuscript, it is evident that the signals generated by the DES are relatively weak, typically in the range of tens of nA with voltages around 1 V. In robot applications, signal fluctuations resulting from electromagnetic interference produced by the robot's motors, drivers, and other components could potentially overshadow the signals generated by the droplets. However, the authors have not described how these issues are addressed in the manuscript.

Reply to reviewer:

We thank the reviewer for the valuable comment.

In the applications, the RC filter circuit (low pass filter: <10 Hz) was used to filter out the high-frequency electromagnetic signals. To understand the signal processing circuit, we have provided the circuit and the processed voltage signal in **Supplementary Fig. 15**. The application in **Fig. 5d** demonstrates that the sensing ability of DES is not affected by electromagnetic interference. In this experiment, a motor and a motor driver were used. In this case, the electrical signal of DES can be effectively perceived by the sensing system, allowing the control system to provide timely feedback. Moreover, we used the droplets in the sliding form to realize the monitoring of droplet slide behavior in our experiments. Droplets in sliding form inevitably produce a lower electrical output than the impinging droplets (Fig. 3f). The impinging droplets can generate a voltage signal of nearly 5 V. The waveform characteristics of sliding droplet signals are very distinguishable and can be effectively identified although the DES outputs a voltage signal of around 1V. The actual application in Fig. 5d proves that the voltage signals generated by DES are sufficient to be recognized by the sensing system, and will not be affected by electromagnetic interference.

Supplementary Fig. 15 | The Architecture of the intelligent monitoring system. (a) DES with 8-channel output. (b) Signal processing circuits and MCU used in the applications. (c) LED trajectory indicator, sliding direction indicator and robotic hand for applications. (d) Voltage signal output from signal processing circuit.

Fig. 5d (i) | Intelligent sense-control system with motor and driver.

7 The authors claim that the designed DES exhibits excellent flexibility. However, the manuscript does not provide information on the electrical output of the DES under deformation or the stability of the signals during repeated deformation processes. Additionally, the application cases presented in the manuscript show that the DES is fixed on a flat surface. In reality, however, the surfaces of robotic hands or human skin are not flat.

Reply to reviewer:

We appreciate the reviewer's valuable comment.

The DES is made of soft material and shows good flexibility and great adaptability on non-flat surfaces such as robot limbs. Based on the reviewer's suggestion, we provided the stability of the signals during repeated deformation processes. As shown in **Supplementary Fig. 9**, the electrical output of DES before and after 1,000 repetitions of 90° bends was performed. It is easy to find that DES has no obvious electrical decay after repeated deformation. **The deformation stability of DES was supplemented in Supplementary Fig. 9 and on page 15 in the revised manuscript.**

Additionally, the reason for attaching the DES to a flat surface in the experiment is to control the experimental parameters such as sliding velocity and direction. In fact, on a non-flat surface, information about the relative droplet motion behavior can also be captured. **Based on the questions raised by the reviewers, we provided relevant validation in the following Figure.** The following Figure shows the sliding path of a droplet on flat and non-flat DES and its multichannel electrical signal. In the case of flat DES (Figure a), the droplet tends to slide along a straight trajectory in the Y-direction, passing sequentially through the four X-electrodes along a single Y₂ electrode. The sliding trajectory reflected by the electrical signal is consistent with the actual path, i.e., X₁, Y₂, X₂, Y₂, X₃, Y₂, X₄. **Furthermore, the droplet sliding behavior on the non-flat DES can also be accurately detected.** When the droplet slides along the Y-direction on the non-flat surface, the trajectory of the droplet is shifted to the neighboring electrode as shown in Figure (b). All these changes and differences are reflected in detail in the acquired multi-channel electrical signals. The results show that the droplet passes sequentially through four X electrodes along the Y₂ and Y₁ electrodes. The corresponding sliding trajectory is X₁, Y₂, X₂, Y₂, X₃, Y₁, X₄. Therefore, the sliding behavior of droplets on both flat DES and non-flat DES can be efficiently perceived.

Supplementary Fig. 9 | Comparison of electrical output of DES before and after 1,000 repetitions of 90° bends.

(a) Droplet sliding on flat DES and (b) sliding on non-flat DES after deformation and the multichannel electrical signals.

Reviewer #2 (Remarks to the Author):

This manuscript reported an electronic skin capable of tracking the two-dimensional dynamic sliding trajectories of water droplets. The electronic skin consists of a nonwoven fabric substrate, flexible interlaced electrode, and a negative triboelectric film, showing good flexibility and conformity to non-planar surfaces such as robot limbs. The key novelty of the electronic skin lies in the interlaced electrode design which enables the detection of temporal differences among the electrical signals induced by a sliding droplet and thereby allows real-time record of the droplet's motion direction. Combined with the microcontroller unit for signal processing, the electronic skin can be further utilized for intelligent closed-loop control of robots to prevent liquid droplet leakages. However, this work primarily focused on tracking the sliding motion of the individual droplet and did not address the detection of more common scenarios involving multiple droplets sliding simultaneously. The practical implementation of this work seems to be limited due to its sole reliance on water droplets as the sensing object. Additionally, the manuscript does not provide sufficient information regarding the stability of the electronic skin's performance. I also recommend that the authors carefully reexamine their calculation results for droplet acceleration, specifically focusing on cases where the calculated values appear unreasonably high. Thus, this manuscript could be considered for publication after the authors address the

following technical comments.

Thanks a lot for reviewer's comment for improving our manuscripts. Based on the comments and suggestions, we provided the electrical output of DES for monitoring multiple droplets **on page 19** in the revised manuscript and confirmed that DES can effectively detect the sliding behavior of multiple droplets. Additionally, we supplemented the continued stability and long-term stability of DES **on page 15** in the revised manuscript to confirm the reliability. We thank you for the reminder that the calculation for droplet acceleration, and we have corrected it **on page 13** in the revised manuscript. **The following is our point-to-point response:**

1. The illustration of working mechanism in Fig. 2a-d appears to be elusive. It is not easy to understand the process of electrical induction and the charge transfer induced by droplet sliding. I recommend the authors to make additional explanations in both schematics and discussions.

Reply to reviewer:

Thanks a lot for reviewer's comment.

We have supplemented the working mechanism to the revised main manuscript (on page 10) and replaced the Fig. 2b. The working mechanism is described as:

Fig. 2b | Schematic of the working mechanism of DES.

“Initially, due to the strong electronegativity and the generated negative charges on the FEP triboelectric layer, the electrodes are induced and exhibit an equal number of positive charges, achieving an electrostatic equilibrium. As the positively charged droplets slide toward the first electrode, the previous equilibrium is broken and a new electrostatic equilibrium between the FEP surface and the water droplet is formed (Fig. 2b-i). To neutralize the excess positive charge in the electrodes, a stream of electrons flows from the ground to the electrodes through an external circuit, creating an electrical signal. As depicted in Fig. 2b-ii, once the water droplet approaches the second electrode, the first electrode induces an opposite current, and the electron flow is induced from the ground to the second electrode.”

2. The authors chose the tap water, rain water, and DI water as the research objects

in the manuscript. What is the sensing capability of the electronic skin for aqueous solutions (such as brine, acid or alkali) and organic liquids (such as ethanol or acetone)? They are more common in the environments requesting the robot workers, for example, workshops, chemical laboratories, and smart medical rooms.

Reply to reviewer:

We thank the reviewer for the careful review and comment.

Based on the reviewers' comments and suggestions, we have supplemented the data and description on DES monitoring more types of liquid (DI water, tap water, rainwater, 0.1 M NaCl, 0.1 M NaOH, 0.1 M HCl solutions, and seawater) as shown in Fig. 3g. The result shows that DI water generates the highest voltage, followed by the rains and tap water. However, the other acid/alkali solutions exhibit lower electrical signals, especially the lowest for seawater. The most significant reason why DES can clearly distinguish between various kinds of liquid should be the different triboelectric charges generated by the liquid and the DES surface. Specifically, DI water is pure and does not contain excess Ca^{2+} , Mg^{2+} , Na^+ , and K^+ , as well as acid ions that avoid the negative effects of charge-shielding, and therefore generate the highest electrical output. The charge-shielding of high-concentration ions in other kinds of liquid, especially seawater, reduces the charge density on the DES surface, resulting in a lower electrical output. This finding proves that DES can be used to effectively monitor a wide range of liquid. This content was supplemented on page 14 of the revised manuscript as:

“Furthermore, different liquid types, such as deionized water (DI water), tap water, rainwater, 0.1 M NaCl, 0.1 M NaOH, 0.1 M HCl solutions and seawater, are used as droplets for the investigation. The electrical output shows that DI water outputs the highest voltage, followed by the rains and tap water (Fig. 3g). However, the other acid/alkali solutions and brines exhibited lower electrical signals, especially the lowest for seawater. The most significant reason that DES can clearly distinguish between various kinds of liquid should be the different triboelectric charges that these liquids generate with the DES surface. Specifically, DI water is pure and does not contain excess Ca^{2+} , Mg^{2+} , Na^+ , and K^+ , as well as acid ions that avoid the negative effects of charge-shielding, and therefore generate the highest electrical output. The charge-shielding of high concentration ions in other liquid, especially the seawater, reduces the charge density on the DES surface, resulting in a lower electrical output.”

Fig. 3g | Electrical signals generated by different liquid types.

3. The electronic skin can effectively sense the dynamic parameters of the single droplet. What is the electronic skin's performance in detecting multiple droplets?

Reply to reviewer:

We thank the reviewer for this constructive comment, and we agree with reviewer that the DES may contact multiple droplets.

In this regard, we have measured and confirmed that DES can accurately detect multiple droplets. As shown in Fig. 4i, j, three adjacent droplets are sliding on the DES surface at the same time and passing through the corresponding electrodes in turn. The order of the electrodes passed by the three droplets and their sequence of the electrical signals are labeled as 1-2-3-4-5-6-7; a-b-c-d-e-f-g; i-ii-iii-iv-v-vi-vii. The multichannel electrical signals demonstrate that the sequence and position of the electrical signals generated correspond exactly to the actual path of the droplet sliding. This demonstration strongly confirms that the motion behavior of these multiple droplets is still accurately detected by DES.

Fig. 4i, j | Schematic of multiple droplets sliding and their multichannel electrical signals.

This content was supplemented on page 19 in the revised manuscript as:

“In reality, DES may be exposed to more than one droplet at a time, so the DES must have the ability to sense the motion of multiple droplets simultaneously. As shown in Fig. 4i, j, three droplets are sliding on the DES surface at the same time and passing through the corresponding electrodes in turn. The order of the electrodes passed by the three droplets and their sequence of the electrical signals are labeled as 1-2-3-4-5-6-7, a-b-c-d-e-f-g, and i-ii-iii-iv-v-vi-vii, respectively. The multichannel electrical signals demonstrate that the sequence and position of the electrical signals generated correspond exactly to the actual path of the droplet sliding. This demonstration strongly confirms that the motion behavior of these multiple droplets is still accurately detected by DES.”

4. The performance stability of the electronic skin is rarely discussed in the manuscript. Many works have mentioned that the performance of triboelectric films is vulnerable to the ions (10.1002/adma. 201905696; 10.1016/j.xinn. 2022.100301). Even in DI water, there exist hydrogen ions and hydroxide ions. I am curious about whether ions would undermine the stability of the electronic

skin's performance and cause the transition of the sensing signals when the electronic skin experiences long term exposure to in water. In addition, would other environmental conditions, including humidity, temperature and air pressure, affect the signals?

Reply to reviewer:

We appreciate the reviewer's valuable comment, and we supplemented the continued stability and long-term stability of DES, as shown in Fig. 3 h, i. The electrical output signal was measured continuously for 40 minutes. There was a slight degradation in electrical performance at 10 minutes, which may be due to the negative effects of charge-shielding caused by ions. No degradation in electrical output was observed during the next 30 minutes, although there were slight fluctuations during the measurement. Compared to the initial electrical performance, the electrical output of the DES after one year of indoor storage shows only negligible degradation.

Fig. 3 h, i | Continuous electrical output stability and long-term stability of DES. This content was supplemented on page 15 in the revised manuscript as:

“In addition to the sensitivity of the DES, good stability is the essential performance to ensure that it can work for a long time in complex environments. Figures 3 h and i show the continued stability and long-term stability of DES. The electrical output signal was measured continuously for 40 minutes. There was a slight degradation in electrical output at 10 minutes, which may be due to the negative effects of charge-shielding caused by ions. No degradation in electrical performance was observed during the next 30 minutes, although there were slight fluctuations during the measurement. Compared to the initial electrical performance, the electrical output of the DES after one year of indoor storage shows only negligible degradation.”

Unlike solid-solid TENG, which can be severely affected by air humidity, this liquid-solid TENG device generates electrical signals by contacting liquid droplets. The ambient humidity of the surface after DES contact with droplets is much higher than that of the air, so the effect of the air humidity is negligible. Generally, as long as the output electrical signal is sufficient for system identification, the influence of external air factors can be ignored.

5. On page 10, the authors claimed that the acceleration of the sliding droplet could be calculated as up to 11.87 m/s^2 via the signal treatment. This value has been far higher than the gravitational acceleration of 9.8 m/s^2 . The result is confusing because the gravitation should be the main driving force to push the droplet sliding and no other force could further accelerate the sliding droplet.

Reply to reviewer:

Thanks for the careful review and kind reminder.

We apologize for the miscalculation because of the misuse of wrong data. We performed new measurements at 45, 55, 65 and 75° using a 13x13 cm DES device. The Δt are 0.36, 0.32, 0.27 and 0.24 s, respectively. The sliding distance of the droplet from contacting to leaving the electrode is approximately 13.5 cm. The average velocity (v) and acceleration (a) of the sliding droplets can be derived from the relationship between

the displacement and the moving time through the following equation of $v = \Delta l / \Delta t$ and $a = 2\Delta l / \Delta t^2$, where Δl is the distance between four electrodes, Δt is the time difference between the four signal peaks.

We have carefully calculated them as follows:

Surface angle (°)	45	55	65	75
Δt (s)	0.36	0.32	0.27	0.24
v (m/s)	0.38	0.42	0.50	0.56
a (m/s ²)	2.08	2.64	3.70	4.68

We have gone through the whole text and corrected it **on page 13** in the revised manuscript. “As the surface angle increases from 45° to 75°, the electrical output increases significantly from 0.85 V to 1.03 V. Under the above four surface angles, the calculated average velocities of water droplets sliding are 0.38, 0.42, 0.50, and 0.56 m/s, and the accelerations are 2.08, 2.64, 3.70, and 4.68 m/s², respectively.”

6. The sensing experiments appear to set up on the static electronic skin. What about the sensing result for the electronic skin attached on a moving robotic arm?

Reply to reviewer:

We thank a lot for the reviewer’s careful comment.

Essentially, DES monitors the relative motion of droplets. Taking the DES surface as a reference, the droplet is always seen to move in a certain direction relative to the DES surface, regardless of whether the DES is in a moving or static state. In this case, the essential relative motion of the droplets is similar to the study in **Fig. 4**. When a droplet hits the DES surface from a distance, the movement of the droplet will transform into sliding (**Fig. 3f**). A special case is when a droplet hits the DES surface, and the droplet is bounced away from the DES. In this way, only a few signal peaks can be collected, but no droplet sliding signal.

7. To better the significance of this work, a comprehensive overview of the existing state-of-the-art droplet sensing and energy harvesting technologies should be supplemented in the introduction section of the manuscript. For example, 1. doi.org/10.1002/dro2.77.

Reply to reviewer:

Thank the reviewer for this suggestion.

The paper suggested by the reviewer was cited as **ref. 36 on page 4** for explaining that the LS-TENG-based droplet sensors are an ideal candidate for droplet electronic skin.

Based on the reviewers' suggestions, we have supplemented recent advances in triboelectric droplet sensing technology in the Introduction Section (on page 4) to fully explain the significance of this work.

“For example, some LS-TENG with the top electrode is developed to detect droplet parameters, such as the dripping height, dripping frequency, and other liquid properties by analyzing the unique characteristics of the output pulse of the sliding droplet. Researchers also designed a cone-shaped interdigital electrode based TENG by employing an interdigital electrode to increase the electrical output. This device can be used as a sensor to sensitively monitor the surface angle of the device, droplet volume rate, velocity, and droplet frequency. The LS-TENG can also be used as liquid droplet counters. Based on a flow-through front surface electrode and metal-dielectric junction, a self-powered water drop counter was proposed. Each droplet signal generated by the drop counter can flash the LED and then be detected by a silicon photodetector, achieving precise detection for liquid drops. Besides, by utilizing the triboelectric effect of liquid droplets/bubbles inside a thin tube, a self-powered microfluidic sensor based on LS-TENG was developed for real-time liquid/gas flow monitoring. The flow rate and flow volume can be effectively derived based on the interval time between signals and the number of signals in a certain interval. There is also a single electrode LS-TENG with a p-type silicon surface proposed for droplet leakage identification and detection. This device is sensitive to leaking liquid and could qualitatively analyze liquid leakage rates. The fiber-based self-powered droplet sensor we proposed earlier can be widely used to monitor the fog intensity, the amount/frequency of raindrops, and water leakages. It should be noted that these droplet sensors mentioned above are only able to monitor the simple or **single-directional (one-dimensional)** dynamic sliding information (i.e. dripping frequency, sliding velocity and acceleration) of the droplet. The recently proposed droplet sensor consists of four LS-TENG units in a single-electrode mode that can monitor the position of dripping droplets at a **multiple-directional (two-dimensional)** level, which may be a breakthrough in the perception dimension. However, due to limitations in the design of the device structure, their perceptual capacity still needs to be improved, and there is obvious crosstalk among the multichannel signals. **The ideal droplet e-skin should not only determine simple features for droplet contact but also perceive the more complex two-dimensional information such as sliding directions, positions, and even trajectories.** To this end, there is an urgent need to develop a droplet e-skin to fully perceive the dynamic information of liquid droplets and improve the augmented perception and autonomous regulation abilities of intelligent robotic devices.”

8. The authors should provide more detailed discussions about the device fabrication. For example, how to make the cross shape of the electrode? How to firmly attach the FEP film onto the electrode surface? Is there adhesive between them? It is hard to find something there in the SEM image (Fig. 1d).

Reply to reviewer:

We thank a lot for the reviewer's careful review and suggestion.

Based on the recommendations of the reviewers, we have provided detailed discussions

about the device fabrication on page 24 of the revised manuscript. The conductive fabrics were processed into a special cross-shaped structure with a length of 40 mm by **template cutting method**. The branched electrode units are made of double-sided adhesive copper-nickel conductive fabric. Thus, the surface of electrode networks is adhesive. Subsequently, the fluorinated ethylene propylene copolymer as a negative triboelectric layer was attached to the adhesive surface of electrodes to obtain the designed DES. **The detailed discussions about the device fabrication on page 24 in the revised manuscript are as:**

“The DES consists of substrate fabric, electrode network, and negative triboelectric layer. Before the fabrication of DES, non-woven substrate fabric was immersed in ethanol for 30 min, and then washed with deionized water and dried. **The branched electrode units are made of double-sided adhesive copper-nickel conductive fabric. The conductive fabrics were processed into a special cross-shaped structure with a length of 40 mm by template cutting method.** Before attaching the electrodes, we first insert a metal wire in the correct position through the interior of the substrate to form an overpass connection together with the wires on the substrate surface. Subsequently, as shown in Supplementary Fig. 1. the row electrode units were first attached to the fabric substrates with the designed rules as X-electrodes. Then the column electrode tapes were attached sequentially as Y-electrodes, which formed a co-layered X-Y electrode network. The continuous electrode units are connected to each other by the overpass connection method. **Finally, the fluorinated ethylene propylene copolymer as a negative triboelectric layer was attached to the adhesive surface of electrodes to obtain the designed DES (13×13 cm).** In addition, other sizes of DES (10×10 cm and 7×7 cm) were prepared to suit a variety of application scenarios.”

Supplementary Fig. 1 | Schematic diagram of the preparation process of DES.

9. The contact hysteresis of the FEP surface should be provided to characterize the droplet sliding dynamic.

Reply to reviewer:

Thanks a lot for the reviewer’s careful review.

Based on reviewer's suggestion, we provided the contact hysteresis of the FEP surface as follow:

(a) Schematic diagram of force analysis of drop pinning on the inclined prepared FEP surface. (b) Contact hysteresis angles of droplets on different inclined FEP surfaces. (c) The sliding state of a droplet on a 65° inclined FEP surface.

Figure (a) illustrates the schematic of the pinning force on a droplet on an inclined surface. The dynamics of the droplet can be analyzed based on equations

$$F_{pinning} = L\gamma_{LA}(\cos \theta_r - \cos \theta_a) \quad \text{and} \quad ma = mg \cdot \sin \alpha - F_{pinning}$$

where L is the contact length of the water droplet; γ_{LA} is the surface tension of water-air; and θ_a and θ_r represent the advancing contact angle and receding contact angle; mg is the gravitational force; ma is the sliding force. The sliding accelerations obtained from the contact hysteresis angles in Figure (b) and the two equations are 3.48, 4.21, 4.79, and 4.93 m/s^2 , respectively. In comparison, the above results are larger than the acceleration deduced from the electrical signal in Fig. 3c (2.08, 2.64, 3.70, and 4.68 m/s^2). The reason for this difference may be that the acceleration of the droplet gradually decreases during the sliding process. Specifically, it can be found from Figure (c) that the shape of the droplet is unstable, and its length is gradually elongated during sliding. Thus the sliding state of the droplet in Figure (b) is captured in the early stage of the sliding process, because the shape of the droplet is irregular in the later sliding stage. In other words, the sliding acceleration we calculate from the contact angle is the instantaneous acceleration in the early sliding stage. As the droplet shape becomes longer, the sliding resistance (pinning force) increases and therefore the acceleration decreases (the sliding speed continues to increase). Moreover, the acceleration calculated by multi-channel electrical signals is the average acceleration and is therefore lower than the above instantaneous acceleration. In summary, the changing rules of instantaneous acceleration and average acceleration at different tilt angles are consistent.

10. The signal treatment methods used for analyzing and sorting the droplet-induced multiple signals, particularly for cases depicted in Fig. 4b and e, are indeed crucial and require detailed discussion in the Method.

Reply to reviewer:

Thanks a lot for reviewer's comment and suggestion.

Based on reviewer's suggestion, we have supplemented the method for analyzing multi-channel signals in **Method on page 25** in the revised manuscript as:

“For analyzing droplet trajectories from multichannel signals. When analyzing the trajectory of the droplet, the signal peaks should first be labeled with the correct sequence numbers based on the order in which the peaks appear in the multichannel signal. Then, read the electrode number corresponding to the electrical signal according to the labeled serial number. Finally, list the corresponding electrode numbers in chronological order to obtain the specific trajectory of the droplet.”

11. The results of sensing oblique and loop sliding should be arranged as main figures instead of the Supplementary figures because they are more convincing. The authors should provide more actual photos to introduce the close-loop control system and related experiments given that Fig. 5 is currently filled with schematics.

Reply to reviewer:

We thank you a lot for reviewer's valuable comments and suggestions.

(1) Based on reviewer's suggestion, we have arranged the results of sensing oblique and loop sliding in main figures **on pages 18-19** in the revised manuscript.

(2) In addition, to illustrate the closed-loop control system, we have supplemented more actual photos in **Fig. 5d, Supplementary Fig. 17**, and more descriptions **on page 23** in the revised manuscript as:

“The system consists of a piece of DES, an MCU equipped with the preprocessing circuit, a motor driver, a stepper motor, and a hand model (Fig 5d (i)). In detail, as shown in Fig 5d (ii) and Supplementary Fig. 17, the DES-equipped robotic hand senses the liquid leakage in a specific direction from the cup and sends the analog signal to the MCU. The MCU determines the direction of the leaking droplets based on logical analysis of signal characteristics and immediately sends a digital signal command to the motor driver of the robotic hand to adjust the cup to the correct state (Supplementary Movie 5). During this process, the MCU can simultaneously send control commands to the driver and receive constant feedback from the DES. The MCU will only stop sending commands to the driver when the hand has been adjusted to the correct position, thus realizing closed-loop control. This demonstration can not only realize the perception-feedback of droplet sliding behavior, but more importantly, but also achieve precise control like human nervous regulations, showing great value and potential for intelligent robotics applications.”

Fig. 5d | (i) Intelligent sense-control system and (ii) the closed-loop control mechanism used for an intelligent robot in a smart restaurant.

Supplementary Fig. 17 | Components and mechanisms of closed-loop control systems.

REVIEWERS' COMMENTS

Reviewer #1 (Remarks to the Author):

The authors have thoroughly addressed the issues I raised previously, incorporating a substantial amount of additional content based on my comments. Their comprehensive revisions have effectively resolved all my concerns, and I have no further questions. I recommend that the paper be accepted for publication.

Reviewer #2 (Remarks to the Author):

I am happy with all the changes made by the authors. The authors have further improved the merit and quality of this manuscript.

Dear Reviewers:

Thank you for your letter and for the reviewers' comments concerning our manuscript entitled "Bionic E-skin with Precise Multi-Directional Droplet Sliding Sensing for Enhanced Robotic Perception". Your comments are all valuable and very helpful for revising and improving our paper, as well as the important guiding significance to our research.

REVIEWERS' COMMENTS

Reviewer #1 (Remarks to the Author):

The authors have thoroughly addressed the issues I raised previously, incorporating a substantial amount of additional content based on my comments. Their comprehensive revisions have effectively resolved all my concerns, and I have no further questions. I recommend that the paper be accepted for publication.

Response:

We thank a lot for reviewer's acceptance to our manuscript. We will keep spending enough effort to make more contributions.

Reviewer #2 (Remarks to the Author):

I am happy with all the changes made by the authors. The authors have further improved the merit and quality of this manuscript.

Response:

Thank you very much for your positive comments on the manuscript.